# Dynamic Transcriptional Landscape of Grass Carp (*Ctenopharyngodon idella*) Reveals Key Transcriptional Features Involved in Fish Development

**DOI:** 10.3390/ijms231911547

**Published:** 2022-09-30

**Authors:** You Duan, Qiangxiang Zhang, Yanxin Jiang, Wanting Zhang, Yingyin Cheng, Mijuan Shi, Xiao-Qin Xia

**Affiliations:** 1Institute of Hydrobiology, Chinese Academy of Sciences, Wuhan 430072, China; 2University of Chinese Academy of Sciences, Beijing 100049, China; 3The Innovative Academy of Seed Design, Chinese Academy of Sciences, Beijing 100101, China

**Keywords:** grass carp, development, zygotic activation, single-molecule real-time RNA sequencing, retained intron, LncRNA

## Abstract

A high-quality baseline transcriptome is a valuable resource for developmental research as well as a useful reference for other studies. We gathered 41 samples representing 11 tissues/organs from 22 important developmental time points within 197 days of fertilization of grass carp eggs in order to systematically examine the role of lncRNAs and alternative splicing in fish development. We created a high-quality grass carp baseline transcriptome with a completeness of up to 93.98 percent by combining strand-specific RNA sequencing and single-molecule real-time RNA sequencing technologies, and we obtained temporal expression profiles of 33,055 genes and 77,582 transcripts during development and tissue differentiation. A family of short interspersed elements was preferentially expressed at the early stage of zygotic activation in grass carp, and its possible regulatory components were discovered through analysis. Additionally, after thoroughly analyzing alternative splicing events, we discovered that retained intron (RI) alternative splicing events change significantly in both zygotic activation and tissue differentiation. During zygotic activation, we also revealed the precise regulatory characteristics of the underlying functional RI events.

## 1. Introduction

The absolute star of RNA study has been mRNA for a very long time. However, over the past 20 years, human understanding of RNA has substantially increased, and an increasing number of RNA kinds, functions, and regulatory mechanisms have been identified. The general functions of long non-coding RNAs (lncRNAs) and alternative splicing in biological processes are the most fascinating.

A family of RNA molecules known as lncRNAs are those that are longer than 200 bp and do not encode proteins. They can take part in a variety of life activities, including gene expression regulation, cell differentiation and development regulation, disease occurrence, and other biological processes [1,2,3]. Expressed retrotransposons, for example, are selectively active during the process of mammalian zygotic gene activation (the process of epigenetic reprogramming) [4,5,6,7].

Alternative splicing is the mechanism by which cells generate numerous mature mRNA products from a single pre-mRNA. It is a common mechanism in eukaryotes and is involved in a variety of biological processes, including development, cell and tissue differentiation, and disease incidence [8,9,10]. Retained intron (RI) is one sort of alternative splicing pattern in which alternative introns are not spliced out but are retained in mature mRNA.

RI is the most common type of alternative splicing event in plants and fungi, and it has been extensively researched [11,12]. It has also been gradually demonstrated to be functional in animals [13]. RI may regulate gene expression via a variety of ways, including nonsense-mediated mRNA decay (NMD), blocking mRNA export from the nucleus, and creating novel gene products. It is vital for plant development and stress resistance, and it has also been linked to a number of human disorders [13,14].

RNA-seq technology based on next-generation sequencing considerably simplifies transcriptome research. Because it does not rely on poly(A) tails for library construction, Illumina strand-specific RNA sequencing (ssRNA-seq) technology is frequently employed in systematic identification and expression profiling of lncRNAs [15,16,17]. Due to the fact that it provides specific benefits in the identification of transcript structure as a result of its lengthy sequencing reads, single-molecular real-time sequencing (SMRT)-based RNA-seq is frequently employed for full-length transcript assembly and alternative splicing analyses [18,19].

These sequencing technologies have made it much easier to construct a baseline transcriptome in a number of different species. As a result, our knowledge regarding the function of various RNAs and the alternative splicing events they participate in during the course of development has been significantly expanded [20,21,22,23,24,25,26,27]. Research in developmental biology can benefit from using fish embryo since it is simple to observe and manipulate and is a suitable material to study. If we carry out expression profiling research on a number of various time periods, it will unquestionably assist us in gaining a deeper understanding of the molecular control mechanism of ontogeny and differentiation at the transcriptional level.

Single-molecular real-time (SMRT)-based RNA-seq has been used a lot in fish studies to look at alternative splicing events. However, these studies have mostly focused on transcriptome assembly [28,29,30,31] and specific study subjects [32,33,34,35,36,37,38], and they have not looked at how alternatively spliced transcripts change in expression over time in a systematic way.

At the moment, most research on fish baseline expression is done on zebrafish, the most famous model fish. RNA microarray technology was used by Mathavan et al. in 2005 to look at the expression of 14,904 genes at 12 different times, from when the eggs were still unfertilized to when they were two days old [39]. Later, when RNA-seq technology advanced, the dynamics of gene expression throughout zebrafish development were investigated in more depth [40,41,42,43,44]. In comparison, studies on other fish species are not only scarce, but also of poor quality (with fewer time points). For example, only three samples encompassing three developmental time periods were gathered in a study of the Siberian sturgeon (*Acipenser baeri*) [45]. No more than five samples were collected per species in a comparative transcriptome investigation of six cichlid fishes [46]. Only six time points were explored in the bighead (*Hypophthalmichthys nobilis*) study, primarily including its early stage development [47]. In addition, 7 time points spanning 2 to 10 days post-fertilization (2–10 dpf) were sampled in the investigation of transcriptome dynamics of channel catfish (*Ictalurus punctatus*). However, because developmental morphology was not used as a reference during sampling, crucial time points in embryonic development were not gathered [48].

With an annual output of 5.7 million tons in 2018, accounting for 6.9 percent of the world’s total farmed fish production, grass carp (*Ctenopharyngodon idella*) is the world’s most productive farmed fish kind [49]. Currently, grass carp lncRNA research focuses mostly on the immunological response to infection, environmental stress response, feeding, and growth [50,51,52,53], with no systematic study to describe lncRNAs. The availability of the draft grass carp genome has tremendously aided genetics breeding research and other basic grass carp research [52,54,55,56,57]. However, the draft genome annotation is lacking in information on lncRNAs and alternatively spliced transcripts, making it challenging to meet the demands of increasingly diversified data processing. As a result, a thorough baseline expression investigation on grass carp is required to expand its gene functional annotation while also gaining basic information of gene transcription dynamics in its development and differentiation.

In this study, we used SMRT-based RNA-seq and ssRNA-seq technology to systematically examine the essential transcriptional events in grass carp embryonic development to differentiated tissues. We collected 41 samples of 11 grass carp tissues at 14 time periods, built a transcriptome with a completeness of up to 93.98 percent (BUSCO), enhanced grass carp genome annotation, and recreated the entire grass carp development and differentiation landscape at the transcriptional level. We discovered a SINE family and its motif-related transcripts with unique expression patterns, and we detailed the specificity of RI alternative splicing events in embryonic development and tissue differentiation, as well as their possible regulatory elements.

## 2. Results

### 2.1. Transcriptome Sequencing and Assembly

Given the spatiotemporal specificity of gene expression, we sampled as many time points and tissue types as possible. We obtained 41 samples for transcriptome sequencing in a whole sibling population of grass carp, from fertilized egg to 197 days following fertilization, covering 22 time periods and 11 tissues (embryos are considered one tissue) (Table 1). These developmental stages are separated into four stages: zygotic genome activation (ZGA, T2–T8), gastrula to hatching out (GH, T9–T16), larvae (T17–T20), and juveniles.

After sequencing, we were able to extract a total of 621 Gb of Illumina short reads and 87 Gb of PacBio long reads. Following assembly (shown in Figure 1A), we were able to get a total of 33,055 genes and 77,582 transcripts. In contrast to the annotation results of the grass carp genome (gc.v1) [54], the grass carp transcriptome (gc.v2) includes a greater number of alternative splicing events as well as more annotations for non-coding RNA (Figure 1B,C, Appendix A).

Sequences in gc.v2 (3281 bp, median) are much longer than in gc.v1 (1642 bp) and are longer than the sequences both in loach (*Misgurnus anguillicaudatus*) transcriptome (3054) [28] and in *Onychostoma brevibarba* transcriptome (3054) [29], which were sequenced using SMRT technology. In addition to this, it was significantly longer than the RNAs that were found in the zebrafish transcriptome (2103) (GRCz11) (Appendix A).

The completeness of the gc.v2 transcriptome reaches 93.98 percent, which is better than that of the loach (*Misgurnus anguillicaudatus*) and the *Onychostoma brevibarba*, which are also based on SMRT sequencing; the only other transcriptome completeness that is lower is that of the zebrafish (GRCz11) (Figure 1D). When the SMRT transcripts that could not be matched to the grass carp genome are taken into consideration, the level of completeness of our assembled transcriptome is remarkably similar to that of the zebrafish transcriptome: a difference of 1.72 percent, or 79 out of 4584 total transcripts.

### 2.2. Overview of Grass Carp Development and Tissue Differentiation

#### 2.2.1. The Developmental Trajectory of Grass Carp

The expression levels were used in the principal component analysis (PCA) analysis of all 41 samples, and it was discovered that three-dimensional PCA had the ability to recreate the developmental trajectory and tissue differentiation scenarios of the samples (Figure 2). The principal component 1 (PC1) is able to effectively differentiate embryo from differentiated tissues, and the relationship between the larva fish (T17–T20) within one month of hatching is closer to differentiated tissue. On the other hand, the fish that is hatching (T16) is just located between the embryo and the juveniles.

There are a total of 4428 transcripts that are associated with PC1. According to the results of the GO enrichment analysis, they were primarily found in the nucleus (the cellular compartment, or CC), participated in the biological pathway (BP) that was associated with the metabolism of nucleic acids, and the nucleic acid binding was the most important molecular function that they possessed (Figure 3A). The majority of the considerably enriched KEGG pathways are those that are connected to immunological control and fundamental life regulation (Figure 3B). There are 4068 PC2-related transcripts, with more non-coding RNAs were found than expected (*p* = 5.33 × 10^−38^) (Appendix A). These transcripts were not enriched in any KEGG pathway; GO enrichment analysis shows that they were mainly development-related transcription factors (Appendix A).

There are 4884 PC3-related transcripts, with more mRNAs found than expected (*p* = 2.30 × 10^−9^). GO enrichment analysis shows that they were mostly related to ion transport (Appendix A). The most significant pathways in KEGG enrichment analysis were nicotine addiction, glutamatergic synapse, and neuroactive ligand–receptor interaction (Appendix A).

#### 2.2.2. Fish Brain Is Highly Differentiated

In order to explain changes in tissue composition in grass carp based on expression patterns, we carried out PCA analysis on all tissue samples (Figure 4A). It was discovered that PC1 based on tissue samples was highly comparable to PC3 based on all samples, and that PC2 based on tissue samples was highly similar to PC4 based on all samples. (Appendix A). The differences between the various tissues of grass carp were, on average, substantially greater than those between the several phases of development present in the same tissue (Figure 4A). There was a significant difference between brain tissue and the other tissues, and the majority of this difference was explained by PC1 with an explanatory degree of up to 25 percent. PC1-related genes were mainly enriched in GO terms related to ion channels, ion transport, and synapses (Figure 4B), which clearly reflected the characteristics of metabolic activity in the brain; the most significant pathways of KEGG enrichment analysis were nicotine addiction, glutamatergic synapse, and neuroactive ligand–receptor interaction (Figure 4C).

#### 2.2.3. Muscle Development and Basic Metabolic Regulation Describe Major Differences among Fish Organs

Except for the brain, PC2 and PC3 in tissue PCA analysis described the origin information of tissues jointly (Figure 4D). Muscle (HM) and heart (HH) were grouped together; liver (HL), spleen (HP), gut (HI), and kidney (HK) were grouped together to the left of PC2; and skin and gills were grouped together to the top right.

Significant GO terms in PC2 were largely connected to muscle, while significant KEGG pathways were mostly related to cardiac disease, according to enrichment analysis (Appendix A). PC3-related genes were enriched in GO terms related to enzymes, amino acid metabolism, and organ development, while the significant KEGG pathways were mostly related to immunity (complement and coagulation cascades), regulation (ribosome), and amino acid metabolism (glycine, serine, and threonine metabolism) (Appendix A). Coding RNAs were substantially more abundant in PC2-related genes, whereas non-coding RNAs were more abundant in PC3-related genes (Appendix A).

### 2.3. Expression Analysis of Grass Carp Transposons

#### 2.3.1. Expression Scenery of Grass Carp Transposon

According to the annotation of the grass carp genome, transposable elements (TEs) are widely distributed in the grass carp genome, accounting for 38.06 percent of total genome sequences (Appendix A). Moreover, based on the gc.v2 transcriptome, we discovered that the majority of transcripts are connected with repeat elements (52,686 TE-related transcripts), yet only 64 of these transcripts were classified as transposons (see Section 4). When compared to their distribution in the genome, the discovered expressed transposons were imbalanced (Table 2, Appendix A). For example, the widely dispersed DNA transposons DNA/hAT (23.53 percent) were not found to be expressed, although DNA/TcMar (6.29 percent) were. Long terminal repeats (LTR) were a very prevalent retrotransposon (5.22 percent) in the grass carp genome, but only two of them were found to be expressed. The two most often expressed retrotransposons, long interspersed nuclear element (LINE) and short interspersed nuclear element (SINE), belonged to non-LTR that were relatively infrequent (4.8 percent) in grass carp total TEs. Unclassified SINE, on the other hand, accounted for 64.06 percent (41/64) of all expressed transposons, despite accounting for only 0.6 percent of total TEs. Even more intriguing, all of the discovered expressed SINE transposons are members of rnd-3_family-293.

#### 2.3.2. SINE Element rnd-3_family-293 Is Specifically Expressed during Embryonic Development

The embryonic transcriptome of grass carp largely maintained the content of maternal RNA before the activation of the zygotic genome (T2–T6), and the amount of various RNA molecules was reasonably steady (Figure 5A). Transposons’ expression was rigorously restricted at the T2–T5 (ovum to 64-cell) stage compared to other genes, but was clearly expressed at the T6 stage (256-cell, Figure 5B). At the moment, all transposon elements expressed belonged to rnd-3_family-293. These transcripts were only expressed during embryonic development (T6–T16, Figure 5C), and the patterns of expression were strikingly consistent.

#### 2.3.3. Motif Analysis for Family rnd-3_family-293

The rnd-3_family-293 transposons have a tRNA-derived head sequence, a body with a 65 bp Ceph domain [58], a 70 bp CORE domain [59], and a 116 bp tail designated as unknown domain (Figure 5D). We examined the motifs of the 41 expressed transposons in rnd-3_family-293, and the large majority of these sequences contained motif 1 (41/41) and motif 5 (30/41) (Figure 5E,F), with the two motifs positioned in the upstream region of the tRNA header (Figure 5G).

#### 2.3.4. Sequences Possessing Both Motif 1 and Motif 5 Were Specifically Expressed during Embryonic Development

We analyzed the transcriptome for transcripts related with motif 1 and motif 5 to see if they could be regulatory sequences. There was a strong correlation between the expression patterns of rnd-3_family-293 and those of motif-related transcripts (transcripts containing both motifs) (Figure 5C,H); transcripts containing just motif 1 or only motif 5 did not show this expression pattern (Figure 5I,J). We performed qPCR on eight motif-related genes and eight rnd-3_family-293 genes. CIWT.15163.1 and CIWT.15487 were predicted as tRNAs using Rfam for two of the eight motif-related genes, all of which are lncRNAs (Appendix A). As predicted by high-throughput sequencing, the qPCR data showed that motif-related transcripts and the rnd-3_family-293 transposon had different expression patterns from T6 to T8, with the maximum level of expression occurring at T7 and very low levels at other stages (T2, T5, T20) (Appendix A).

### 2.4. Alternative Splicing Events in Grass Carp

#### 2.4.1. Distributions of Grass Carp Alternative Splicing Events

In alternative splicing analysis, only transcripts supported by SMRT sequencing data were included to ensure the correctness of the results. A total of 5339 alternative splicing events were discovered in our study, belonging to 7 types: alternative 3′ splice site (A3), alternative 5′ splice site (A5), alternative first exon (AF), alternative last exon (AL), mutually exclusive exons (MX), retained intron (RI), and skipping exons (SE) (Figure 6A). Among them, SE (855, 24.72%), AF (501, 14.48%), A5 (664, 19.20%), RI (729, 21.08%), and A3 (604, 17.46%) were the most common type of alternative splicing (Figure 6A,B, Appendix A).

The composition ratio of alternative splicing types of grass carp samples from gastrulation to juvenile stage (GH, larva, juvenile) is close to the overall ratio (all) (Figure 6B). While the composition ratio in the ZA stage samples was distinct, the proportion of RI was much lower than in other stages of development (Figure 6B). However, the compositional ratio of ZA resembled that of ZA.zf (zebrafish ZGA stage) (Figure 6B).

#### 2.4.2. RI Events Changes Dramatically during T6–T7

The Ψ value was used to describe the expression preference for splicing events. When Ψ = 1, all expressed transcripts are form 1; when Ψ = 0, all expressed transcripts are form 2; when Ψ = 0.5, two types of transcripts are equally expressed (Figure 6A, see Methods).

The distribution of Ψ values for AL and AF varied widely across all developmental time series samples (T2–T20), but these difference were not statistically significant due to the small number (<40) of detected alternative splicing events for both events (Appendix A). The distribution of the value of splicing patterns A3, A5, AF, and SE altered from flat to narrow from before to after zygotic genome activation (T2–T6), indicating that the two kinds of alternatively spliced transcripts were expressed more uniformly in the embryos after zygotic activation. The most substantial changes in RI occurred during embryonic development, demonstrating a considerable rise in the mean value of during zygotic activation (Figure 6C). According to the *p*-value distribution of RI, its variation in embryonic phases was also substantially greater than that of other splicing types (Figure 6D). The most significant changes of RI events occurred during T6–T7 (*p* = 5.56 × 10^−37^, the largest outlier in Figure 6D), i.e., from the 256-cell stage to the Dome stage, which corresponds exactly to the full activation of the zygotic genome.

#### 2.4.3. Characteristics of RI Events during T6–T7

Differences calculated by subtracting the Ψ value in T6 (256 cells) from the Ψ value in T7 (dome) for the same splicing event reflects the change in expression preference of splicing events during development. The distribution of differences for RI alternative splicing events was significantly right-skewed compared to other events: that is, in the T7 period, more intron-retained (RI) transcripts were expressed. Based on this distribution, we defined alternative splicing events with a Ψ difference greater than 0.07 as “over” events, and others were named with “normal” events (Figure 7A).

Among the “over” events, 151 genes encountered 184 RI events, with the majority of genes experiencing only one RI event and only 28 experiencing RI events greater than or equal to two times (Figure 7B). Among these genes, intron-retained isoforms tended to have modest levels of expression during T2–T6, but their expression levels began to grow after T7. In contrast, the expression of transcripts with introns spliced out did not vary much (Figure 7C). The three GO terms, nucleic acid binding, cellular metabolic process, and gene expression, were significantly enriched in the “over” event-related transcripts (hence referred to as “over” transcripts), indicating that these three GO terms were the specific functions of the “over” transcripts (Appendix A).

In terms of intron length, GC content, branch-point sequence (BPS sequence), and other factors that may affect intron splicing, we compared alternative introns (introns engaged in RI events) to constitutive introns (introns that are always spliced out). Alternative introns in “over” transcripts were shown to be more likely to be located at the 5′ position of the gene, whereas alternative introns in “regular” event-related transcripts (hereinafter referred to as “normal” transcripts) were found to be more likely to be located in the center of the gene (Figure 7D). However, the positions of the two types of introns were not significantly different (Appendix A). More specifically, the alternative introns of the “over” transcripts were longer than the constitutive introns of the “over” transcripts (*p* = 0.03) and were also longer than the alternative introns of the “normal” transcripts (*p* = 3.04 × 10^−^^9^), while the alternative introns of the “normal” transcripts were shorter than the constitutive introns of the “normal” transcripts (*p* = 1.30 × 10^−34^) (Figure 7E, Appendix A).

Constitutive introns showed a similar GC content distributions for both “over” and “normal” transcripts (Figure 7F, *p* = 0.17, Appendix A). The GC content of alternative introns was higher than that of constitutive introns in both types of transcripts, among which the GC content of alternative introns of “normal” transcripts was the highest and was more dispersed (over: *p* = 2.56 × 10^−^^16^; normal: *p* = 6.56 × 10^−^^39^). However, the distribution of GC content of alternative introns was not significantly different between “normal” and “over” transcripts (*p* = 0.05, Figure 7F, Appendix A).

The likelihood of a splice site being used can be characterized by splice junction strengths [60]. The splice junction strengths of alternative introns were significant weaker than of constitutive introns (5′ splice site, 5′ss: *p* = 2.04 × 10^−44^, 3′ splice site, 3′ss: *p* = 2.49 × 10^−29^, Figure 7G, Appendix A). In the comparison of the splice site strength of introns between “over” and “normal” transcripts, no difference was found in constitutive introns (5′ss: *p* = 0.11, 3′ss: *p* = 0.14), while significant differences were found in alternative introns (5′ss: *p* = 5.47 × 10^−3^, 3′ss: *p* = 0.04). In particular, the splice junction strengths of alternative introns in “normal” transcripts were weaker (Figure 7G, Appendix A).

The BPP software predicted that the BPS sequence of grass carp is NNNYNAN (Y denotes pyrimidine, N denotes any base). The BPS of alternative introns was lower than that of constitutive introns (*p* = 0.05). However, the BPS scores of alternative introns did not differ between “over” and “normal” transcripts (Figure 7H, Appendix A).

#### 2.4.4. Distribution of Splicing Isoforms in Nine Tissues and the Tissue Specificity of RI Alternative Splicing

The number of observed alternative splicing events in the 9 sequenced tissues of juvenile fish was positively correlated with the number of expressed genes (Appendix A). The most expressed genes and alternative splicing events were found in the brain and gills, while the least were found in the liver and muscle. The ratio of alternative splicing events expressed in each tissue was relatively stable; the four categories of most expressed alternative splicing events were A3, A5, RI, and SE (Appendix A).

We conducted pairwise comparisons of the distribution of Ψ values for alternative splicing across juvenile tissues, and the results differed slightly from those in developmental time series samples, as stated above. The results of pairwise tissue comparisons revealed that both RI and SE have highly tissue-specific expression (Figure 6D). Muscle was the most distinct among them in terms of RI alternative splicing events (Appendix A). The expression tendency of RI alternative splicing events differs most significantly between muscle and skin tissues (*p* = 2.95 × 10^−^^23^, Appendix A). Compared with in skin, the “upregulated” RI genes in muscle, which tended to express more intron-preserved isoforms in T7 than in T6 (Ψ(T7)–Ψ(T6) > 0.07, 94/813), were functional related to metabolism, especially energy metabolism. While the “downregulated” RI genes in muscle, which tended to express more intron-excluded isoforms in T7 than in T6 (Ψ(T7)–Ψ(T6) < 0.07, 262/813), were functional related to the regulation of mRNA synthesis. Spleen tissues showed the second-largest difference with muscle in terms of expression tendency of RI alternative splicing genes (*p* = 1.32 × 10^−21^, Appendix A). Compared with in spleen, the “upregulated” RI genes in muscle (79/716) were related to lipid metabolism, while the “downregulated” RI genes in muscle (251/716) were associated with the regulation of amino acids and nucleic acids (Appendix A).

## 3. Discussion

The transcriptome can be evaluated in three ways: (a) comprehensiveness—the fraction of all gene loci that are included; (b) exhaustiveness—the fraction of all transcripts from each locus that are known; and (c) completeness—the fraction of transcript models that cover the entire length of the physical RNA molecule, from start to end [61]. The transcriptome (gc.v2) assembled in this study is the most comprehensive (93.98 percent) compared to other non-model fish transcriptomes [28,32]. When compared to the original grass carp annotation (gc.v1), gc.v2 has a larger transcript/gene ratio (>2), indicating a better annotation of alternative splicing events and a higher level of transcriptome exhaustiveness. We anticipate obtaining a more complete molecular sequence due to the use of SMRT sequencing with longer reads [19,62]. In fact, if the median length of transcripts is used as a metric of transcriptome completeness, gc.v2 (3281 bp) outperforms gc.v1 (1642 bp) (Appendix A).

We created a dynamic picture of the grass carp transcriptome from a larger field of view for the first time under the condition that the transcriptome’s quality is ensured. This is a valuable resource for gaining a better understanding of fish development and tissue differentiation. The differences between samples in developmental time series are primarily represented in changes in the cell cycle, which is the most significant KEGG pathway enriched in the PC1-related transcripts defined in PCA analysis based on all samples. The basic fact related with this discovery is that early embryonic development of fish undergoes fast cell division, with a very short cell cycle, but late development embryos and adult cells have a longer cell cycle [63]. Second, in the same PCA analysis, mid-developmental transition samples differed the most from differentiated tissues and cleavage-stage embryos in terms of PC2 (Figure 2). This aligns nicely with the hourglass model or the ground zero concept: the embryo experiences a process of de-differentiation similar to the induction of somatic cells into pluripotent stem cells from early to middle embryonic development. The expressed genes are more conserved in the mid-embryonic stage, the telomere length is longer, and the entropy of the embryonic structure is lower, indicating that the stemness of the cell is stronger at this time, which is more like the starting point of the organism’s intergenerational rejuvenation [26,64,65].

Although fish are regarded to be lower vertebrates, the brain is highly differentiated and distinct from other organs in terms of tissue differentiation. The linkages between tissue and its origin germ layer are almost evident from the standpoint of development [66], which is reflected in the overall gene expression profiling. Except for highly differentiated brain tissue, tissues developing from the same layers clustered together in the PCA analysis based on all tissue samples. It suggests that tissues coming from the same germ layer may include a large number of genes/transcripts that express similarly: the aggregated muscle tissue (HM) and heart tissue (HH) both contain a large amount of muscle components and are derived primarily from mesoderm; the liver (HL), spleen (HP), intestine (HI), and kidney (HK) are derived primarily from endoderm; and the skin in the upper right is derived from ectoderm. The origin of the gills in fish has been questioned. The gills of jawless vertebrates are thought to originate from the endoderm, while those of jawed vertebrates (cartilaginous and bony fishes) are thought to come from the ectoderm [67]. Using in situ hybridization and destiny mapping techniques, a recent study found that gill filaments in cartilaginous fish arise from the endoderm [68]. Our findings reveal that the gill is placed in the center of the endoderm and ectoderm, highlighting the complexity and specificity of this tissue’s origin.

In vertebrate genomes, transposons and transposon-derived sequences are common [69,70,71]. Transposon elements, which are a major driving force in the production of new genes, are frequently found in expressed genes in the form of repeated sequences [72]. However, because of the potential harm to genome integrity, transposons’ “transposition ability” is normally blocked [73]. This can be seen in the enormous disparities between the genome’s broadly distributed fixed transposons and the modest number of expressed transposons. Transposon DNA/hAT, for example, is thought to be dormant in all species [74], yet it accounts for 12.77 percent of repetitive elements in the grass carp genome and has not been discovered for transcription. However, transposon transcription is not always suppressed. Recent research has revealed that retrotransposons are preferentially activated during mammalian embryonic development, particularly during the process of zygotic gene activation (epigenetic reprogramming) [4,5,6,7,75]. Transposon elements were found to be involved in ZGA events in a study of mouse single-cell RNA sequencing, with LTR associated with the minor wave of zygotic activation and SINE associated with the major wave of zygotic activation, where a large number of essential genes begin to be transcribed [76].

The expressed transposons in grass carp were family-specific and had nothing to do with their abundance on the genome. The transposons of an unclassified SINE family (rnd-3_family-293) were found to be preferentially expressed during embryonic development. More intriguingly, they were activated simultaneously during the 256-cell phase, when the majority of zygotic genes are not yet activated. This shows that fish retrotransposons, like those of mammals, may function during zygotic stimulation of embryonic development. Furthermore, we discovered that the majority of these transposons contained a combination of two patterns (motif 1 and motif 5) in their 5′ tRNA header, and the genes containing the two motifs displayed an expression pattern similar to that of these transposons. Given that SINE transposons are transcribed by RNA polymerase Ⅲ and the binding site for promoter recognition is located within the sequence [77], we believe that this unique expression pattern of rnd-3_family-293 is regulated by a distinct mechanism, which may initiate expression via RNA polymerase Ⅲ recognition of motif 1 and motif 5.

Gene splicing is involved in many biological processes, including development, cell and tissue differentiation, and disease occurrence [9,78]. As a result, systematic mining is very important. We found a total of 5339 alternative splicing events using SMRT RNA-seq technology, with SE (24.72 percent) and RI (21.08 percent) having the largest proportions, which was consistent with the findings of two SMRT-based fish transcriptome studies [31,37]. The percentage of RI events fluctuates substantially during development and is quite low during the grass carp zygotic activation (ZA) stage. This finding corresponds to the transcriptome during zygotic activation in zebrafish [32]. RI splicing events occur far more frequently during other developmental stages (GH, larva, and juvenile). To our knowledge, no other fish species has had such a significant fluctuation in RI events during development. As RI events are demonstrated to be functional, interest in this topic in animals is growing rapidly [13,14,79,80].

In our research, we discovered that RI may be important in both embryonic development and tissue specificity maintenance in fish. The modifications were most pronounced during the maternal to zygotic transition (MZT, 256-cells-Dome), when the number of RI alternative splicing events rose considerably and more isoforms with intron retention were produced. The over genes, the genes with an upregulated Ψ value (Ψ(T7)–Ψ(T6) > 0.07, genes expressing more intron-retained isoforms) during the zygotic activation stage exhibited some interesting features. Compared with the normal genes, the molecular functions of GO terms of over genes were significantly enriched for nucleic acid binding. In general, nucleic acid binding acts as a regulator in life activities.

We are particularly interested in the regulatory aspects of RI events, in addition to their function. In general, alternative introns have lower splicing signals to prevent spliceosome formation and hence remain in the mature RNA molecule. The placement at the 3′ end of the gene, shorter intron length (shorter GT-AG spacing), higher GC content, weaker splice site strength, and lesser BPS sequence strength were the typical weak splicing signals (Table 3). However, the alternative introns we uncovered have some unique characteristics in terms of intron length and splice site strength. In general, alternative introns are short, which makes intron skipping easier during spliceosome formation [80,81]. In a study of chicken embryos, for example, alternative introns were shorter in RI occurrences [79]. Alternative introns in normal events meet this description in our study, whereas introns in over events are longer than constitutive introns. Furthermore, in our investigation, the splicing strength of the alternative introns of the over transcripts was greater than that of the normal transcripts. These distinguishing characteristics suggest that RI alternative splicing events have distinct purposes and meanings during the maternal zygotic transition, as well as a distinct regulatory mechanism.

## 4. Materials and Methods

### 4.1. Sample Collection

Each sample was taken from a population consisting entirely of the offspring of a single set of grass carp parents (bred in Xingfu village, Huanggang, China in 2017). During the process of collecting the samples, the information regarding the samples, such as the amount of time that has passed after fertilization, the type of tissue, and the morphophysiological properties, was recorded (Table 1). The fertilized eggs were continually viewed using a dissecting microscope (OLYMPUS, Tokyo, Japan) after artificial insemination, and the embryonic developmental phases were recognized by reference to the zebrafish development diagram [82].

For the samples taken between T2 and T19, a total of 20 fertilized eggs, embryos, and juveniles were obtained at each time point. The T20 sample consisted of a single fish that was replicated twice for biological analysis. One month after fertilization, sampling was continued every 30 to 60 days. During the sampling process, the grass carp were euthanized with a high dosage of eugenol. This was followed by a quick dissection and the collection of tissue samples, with two to three biological replicates at each time point.

A TRIzol (Invitrogen, Carlsbad, CA, USA) solution was added to the samples, and they were completely crushed in an OSE-Y40 homogenizer (Tiangen, Beijing, China) to release the RNases, which were then inactivated by further contact with a TRIzol solution. The ethics committee gave their blessing to the entire experiment.

### 4.2. Preparation of Sample for Library Construction

The manufacturer’s instructions for utilizing TRIzol reagent were followed when obtaining total RNA. The Qubit^®^ RNA Assay Kit in Qubit^®^ 2.0 Fluorometer (Life Technologies, Gaithersburg, MD, USA) was used to measure the RNA concentration. Using the Agilent 2100 technology, RNA integrity was determined (Agilent Technologies, Santa Clara, CA, USA). It was decided to use samples with RINs greater than 9 for the SMRT-based libraries and greater than 7 for the ssRNA-seq libraries in order to get the most accurate results. Those samples that met the requirements were kept in a −80 °C refrigerator.

### 4.3. Library Construction and Sequencing

#### 4.3.1. ssRNA-Seq

For each tissue sample, a library was generated and sequenced. For each sample point, the qualified RNA from biological replicates were mixed in equal amounts. Three micrograms of RNA was used for each sample in the library construction.

For the first step, the Ribozero^TM^ rRNA Removal Kit (Epicentre, San Antonio, TX, USA) was employed. An Illumina NEBNext Ultra^TM^ Directional RNA Library Prep Kit was used to build the sequencing library, and the manufacturer’s instructions were followed throughout the process. Finally, the Illumina HiSeq 4000 PE System was used to sequence libraries of sufficient quality.

With the help of IlluQC [83], adapter sequences and low-quality sequences that met the Q30 criteria were eliminated. When either the amount of nitrogen (N) in any read exceeded 10 percent of the total number of bases in the read or the amount of bases with a quality rating of 5 or lower exceeded 50 percent, the paired reads were discarded. FastQC and MultiQC were utilized in order to carry out quality information statistics on the aforementioned data [84].

#### 4.3.2. SMRT-Based RNA-Seq

Before being utilized to generate sequencing libraries, the qualifying RNAs that were extracted from embryonic development samples were mixed together in equal amounts. In order to construct the sequencing library using the samples collected after hatching, the RNA samples obtained from the same tissue at various time points were combined in equal amounts to create the tissue samples, and then the tissue samples themselves were combined in equal amounts to create the sequencing library.

The SMARTer^TM^ PCR cDNA Synthesis Kit (PacBio, Menlo Park, CA, USA) was used to do reverse transcription on a mixed RNA sample that was approximately 1 microgram in weight. The PCR product was then utilized in the construction of the SMRTbell library through the utilization of PacBio’s SMRTbell template pre kit 1.0. The SMRTbell template was annealed to the sequencing primer, bound to polymerase, and then sequenced using PacBio’s Sequel Binding Kit 3.0 on the PacBio Sequel platform.

### 4.4. Transcriptome Assembly

#### 4.4.1. Generation Transcriptome for ssRNA-Seq Data and SMRT-Based RNA-Seq Data Separately

We processed the ssRNA-seq data with the classical process: aligned using Hisat2 (2.0.4) before assembled by StringTie (1.3.1c) [85]. The expression level was counted with Salmon (1.4.0) [86].

SMRT sequencing data was analysis with SMRT-link package (5.1.0) and Iso-Seq (Isoform Sequencing) procedure (https://www.pacb.com/support/software-downloads/ accessed on 17 September 2018). First, the command “ccs” in the SMRT-link toolkit was used to obtain circular consensus sequences (CCS); then, “pbtranscript classify” was used to obtain full-length sequences (FLNCs); finally, “pbtranscript cluster” was used to cluster FLNCs to obtain high-quality transcripts. The software gmap (10 June 2019) was used to align high-quality transcripts back to genome, and only results with a mapping rate of over 85% were retained [87]. The “collapse_isoforms_by_sam.py” script in the cDNA_Cupcake package was used to generate a GFF file from the alignment results, with the parameter set as --fq --dun-merge-5-shorter -c 0.95 -i 0.9 (https://github.com/Magdoll/cDNA_Cupcake, accessed on 18 September 2018). Transcripts with full-length CCS support numbers less than 2 and/or degraded from the 5′ side were removed from the generated GFF file.

#### 4.4.2. Merge ssRNA-Seq Transcriptome and SMRT Transcriptome

The ssRNA-seq transcriptome and the SMRT transcriptome were combined, and then gffread (gffread -M -K) was used to eliminate any duplicated transcripts that were found [88]. Using the program “gffcompare”, we compared the merged transcriptome with the ssRNA-seq transcriptome and the SMRT transcriptome in their own right. This allowed us to acquire the supporting information, which indicated whether or not the transcript was detected by both technologies. Salmon (version 1.4.0) was utilized to determine the amount of expression, and transcripts whose values were lower than 1 TPM were eliminated [86]. Two different filtering techniques were carried out to improve the quality of the assembled transcripts even further: Eliminate any transcripts that might have been fragmented. It is quite simple to confuse fragmented mRNA transcripts with newly assembled lncRNAs. In order to locate these transcripts, we first determined the distance between the transcript and the border of the scaffold and then checked to see if the transcript was supported by SMRT data. More than 95 percent of introns were less than 7875 base pairs, as shown in the length distribution of introns in all transcripts (Appendix A). It was decided that the transcripts whose distance from the scaffold edge was less than this value and which were not supported by SMRT data were fragmented transcripts. As a result, these transcripts were removed from the analysis.Remove possible precursor mRNAs. It is anticipated that the RNA-seq technology will discover a high number of mRNA fragments that have not been spliced or are in the process of being spliced. Because these fragments, unlike mature mRNA, exactly correspond to the full continuous region of the genome, it is common practice to confuse them with lengthy single-exon transcripts; nonetheless, they are eventually discovered to be long non-coding RNA. In order to get rid of the negative effects of such transcripts, we made the assumption that if a gene contains multi-exon transcripts as well as single-exon transcripts, the single-exon transcripts are very certainly precursors and should be removed as well.

### 4.5. Annotation of Transcriptome

Two tools, CPAT and CPC2, were utilized to predict the RNA coding ability in order to improve the accuracy of lncRNA prediction. In particular, the CPAT zebrafish model was used for prediction [89]. When both prediction findings from two software were non-coding, a transcript was designated as lncRNA; mRNA when both predictions were coding; and TUCP (transcripts of uncertain coding potential) when the predictions were inconsistent.

If the class of all of a gene’s transcripts is the same, the gene will be designated as the class of its transcripts; otherwise, the gene will be labeled “misc”.

A gene’s functional annotation is assigned based on the annotation result of its longest transcript. The GO function was annotated using blast2go (2.5.0), and the BLAST database was chosen from the nr library [90]. KO annotation was performed using the KAAS system [91] was utilized for KO annotation, and the species used for alignment were human (has), mouse (mmu), zebrafish (dre), common carp (ccar), medaka (ola), and Atlantic salmon (sasa). In KAAS annotation, the bidirectional best hit (BBH) approach was utilized. The RNA categorization (family) was annotated using cmscan (1.1.2) to search the Rfam database (14.1) [92,93].

### 4.6. Completeness of Transcriptome

The loach SMRT-based transcriptome [28] was obtained from the links mentioned in their paper. The SMRT-based transcriptome of *Onychostoma brevibarba* was obtained from the publication’s supplementary material [36]. The completeness of all transcriptomes was determined using BUSCO software (3.1.0) and OrthoDB’s actinopterygii (ray-finned fish) as analytic database [94].

### 4.7. Principal Component Analysis

The “prcomp” function in R was used to perform principal component analysis (PCA). The factoextra package was used to calculate the “variable contributions” of the transcripts to each principal component (PC). The PC-related transcripts were those that were associated with the PC with the highest contribution value. The plotly program was used to create interactive plots. 

### 4.8. Annotation and Analysis of Transposons

#### 4.8.1. Identification, Annotation, and Relationship Analysis of Transposons

Assembled transcripts and transposable elements (TEs) in the grass carp genome (Appendix A) were used to calculate the overlap information [54]. Expressed transposons are transcripts with overlaps of more than 90 percent of repeat elements and 90 percent of the transcript itself. TE-associate transcripts are those whose repeating element overlap exceeds 90% but does not exceed 90% of the transcript’s own overlap. Transposons having N sequences within 20 bp upstream and downstream were specifically targeted for deletion. In order to annotate the structure of short interspersed elements (SINEs), we consulted the SINE Base database [95]. EBI clustal omega [96] was used to analyze the relationships between different sequences.

#### 4.8.2. Motif Analysis

Motif was predicted de novo using MEME (5.3.3) [97]. FIMO (5.3.3) was used to search for motifs in all transcripts with MEME-1 starting coordinates less than 50 and MEME-5 starting coordinates less than 40 [98]. MAST (5.3.3) was used to search and visualize motif combinations [99]. 

### 4.9. Validation Using qPCR

The NEB E. Coli Poly(A) (#M0276S) kit was used to add Poly(A) tail. Genomic DNA removal and first strand synthesis were performed using the HiScript^®^ Ⅲ RT SuperMix kit for qPCR (+gDNAWIper) (Vazyme, China). ChamQ SYBR qPCR Master Mix (Vazyme, China) kit and CFX96 Real-Time PCR Detection System (Bio-Rad, USA) were used to perform RT-qPCR. The RPS27 gene (small subunit ribosomal protein S27e) was selected as the internal reference [100,101]. Annealing conditions of internal reference and target gene primers were recorded (Appendix A).

### 4.10. Alternative Splicing Analysis

The suppa software [102] was used to identify alternative splicing events and calculate Ψ values. The transcripts participating in the event must have a minimum total expression of more than 1 TPM. The following is the formula for determining the value: (1)$$\Psi =\sum_{k\in F_{1}}TPM_{k}/\sum_{j\in F_{1}\cup F_{2}}TPM_{j}$$

*F*_1_ is the set of transcripts spliced following the form 1 in a splicing event, and *F*_2_ is the set of transcripts spliced following the form 2 in the same event. Because only simple splicing events were analyzed (no splicing event combinations were included), there were only two splicing forms for a splicing event. The Wilcoxon signed-rank test was used to compare the distributions of adjacent time points and tissues. All time points from T2 through T20 are considered adjacent. Tissue comparisons were performed using samples from the latest two sampling time points (134 dpf and 197 dpf).

The weblogo online application [103] was used to view the patterns of splice sites. The Maximum Entropy Model (MaxEnt) [104] was used to forecast the splice site strength. The software BPP [105] was used to forecast the branch-point sequence (BPS). The Mann–Whitney *U* test was used to assess distribution differences in splice site strength, BPS scores, intron length, and GC content.

### 4.11. Other Analysis

Fisher’s test was used for KEGG enrichment analysis, while the R package “GOStats” was used for GO enrichment analysis [106]. Other data analysis was performed on the Linux (CentOS 7) platform using R (4.0.2), Python (3.6.11), and other software. The ggplot2 package was used for data visualization. Unless otherwise specified, the default parameters of the bioinformatics program used in this work were used during data processing.

## 5. Conclusions

Through continuous, high-frequency sampling, we obtained a high-quality reference transcriptome and dynamic transcriptional map of grass carp, an important resource for fish developmental study. On this basis, we systematically analyzed the changes of lncRNAs and alternative splicing events in grass carp development.

The zygotic genome activation (ZGA) is an extremely important research topic in fish developmental biology, but its mechanism has not been fully elucidated yet. Based on the reference transcriptome, we discovered a novel family of short interspersed nuclear element and its potential regulatory motifs associated with fish ZGA. Meanwhile, it was found that retained intron (RI) alternative splicing events dramatically change during ZGA, and this type of RI events are different with classical RI events in their regulatory characteristics. In addition, two types of alternative splicing events, RI and skipping exons (SE), are associated with tissue specificity maintenance in fish.

## Figures and Tables

**Figure 1 ijms-23-11547-f001:**
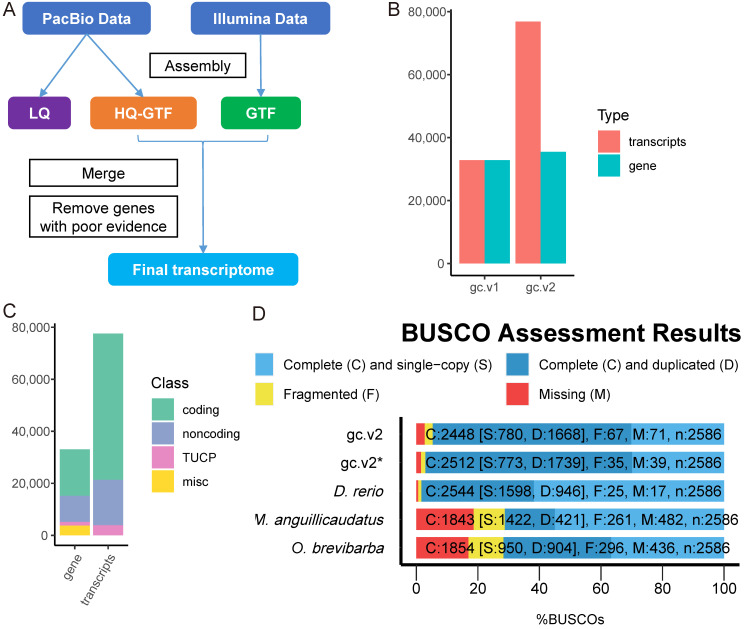
The process of constructing the whole transcriptome of grass carp. (**A**) A schematic representation of the assembly of the transcriptome. (**B**) The gc.v2 genome contains a greater number of transcripts and genes than the gc.v1 genome. The gc.v2 is the transcriptome that was assembled in this study, and the gc.v1 is the annotation information that was found in the draft grass carp genome [54]. (**C**) Statistics on the types of transcripts found in the gc.v2 transcriptome. (**D**) Analysis of the completeness of the gc.v2 transcriptome using BUSCO, which is based on a database of ray-finned fish. The result of calculating gc.v2* was done so under the assumption that all clusters produced by SMRT sequencing are able to generate transcripts based on the grass carp genome. This means that gc.v2* is equal to gc.v2 plus the transcripts that cannot be successfully aligned to the grass carp genome in the SMRT data analysis. *D. rerio* denotes the zebrafish transcriptome (GRCz11); *M. anguillicaudatus* represents the loach transcriptome based on SMRT sequencing [28]; *O. brevibarba* denotes the *Onychostoma abrevibarba* transcriptome based on SMRT data [36].

**Figure 2 ijms-23-11547-f002:**
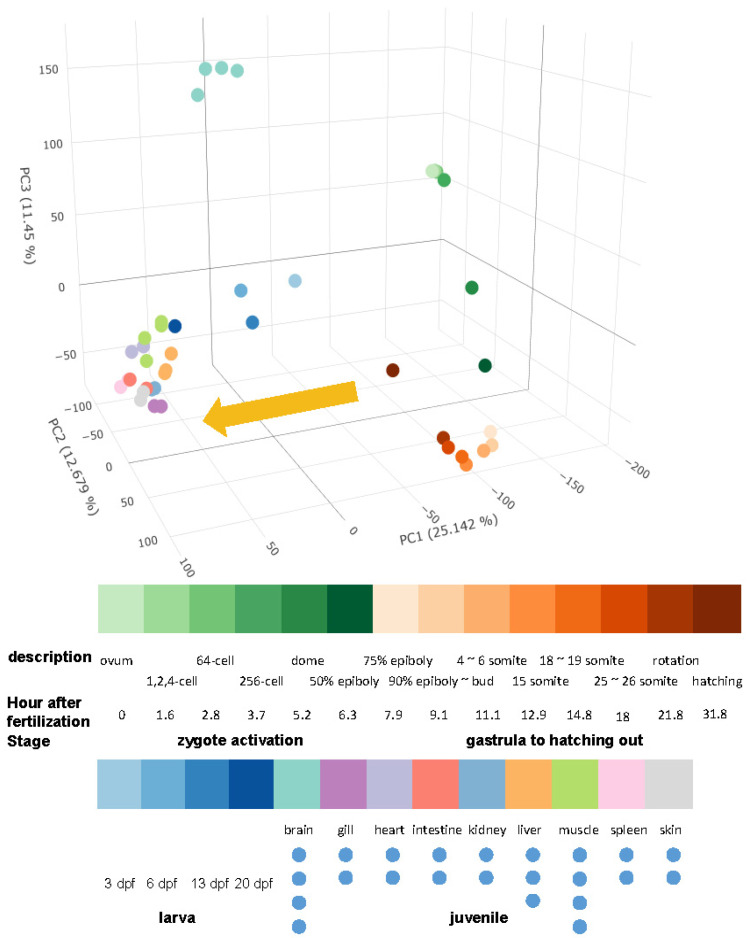
Principal compartment analysis (PCA) performed on all samples. (Top) Three-dimensional PCA diagram, in which the embryo development direction is indicated by the yellow arrow. (Bottom) The color panel and the development information for all samples.

**Figure 3 ijms-23-11547-f003:**
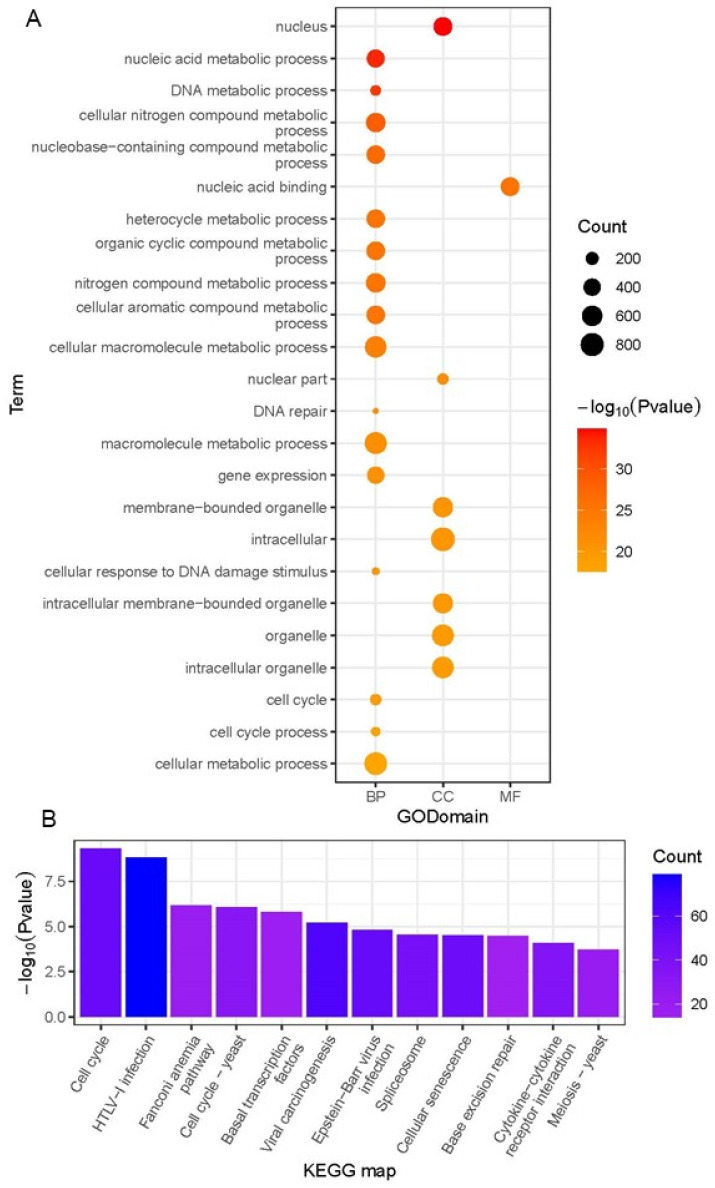
Functional enrichment results of PC1-related genes in all samples. (**A**) GO enrichment results (top 24); (**B**) KEGG enrichment results (top 12).

**Figure 4 ijms-23-11547-f004:**
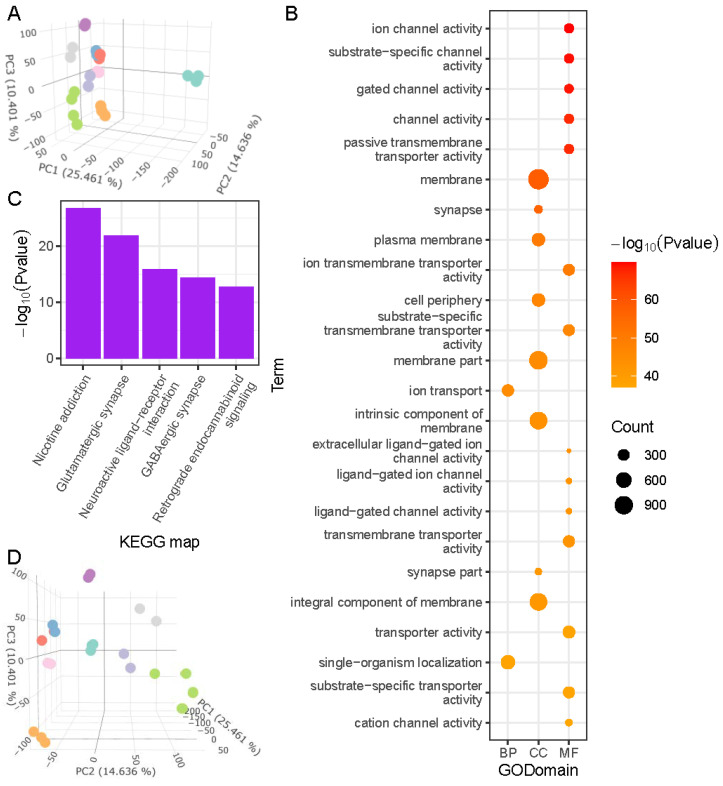
PCA results based on tissue samples. (**A**) Three-dimensional diagram of PCA results (sample colors are shown in Figure 2); (**B**) GO enrichment results of PC1-related genes (top 24); (**C**) KEGG enrichment results of PC1-related genes (top 5); (**D**) PC2 and PC3 (sample colors are shown in Figure 2).

**Figure 5 ijms-23-11547-f005:**
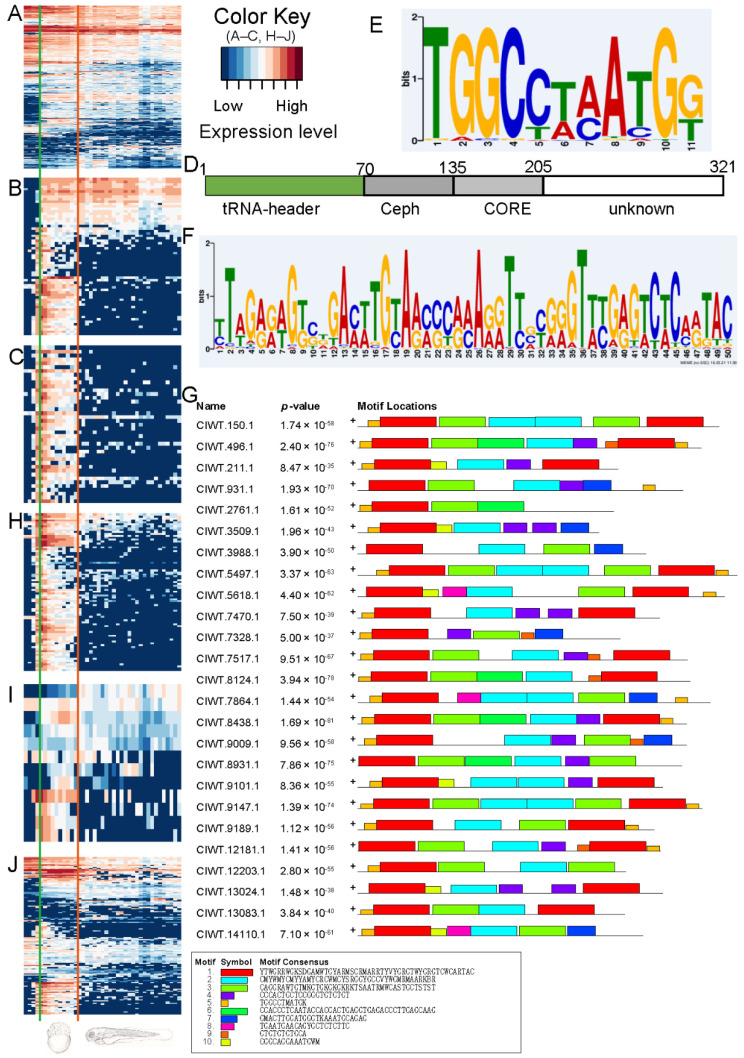
The grass carp transposons that have been activated. (**A**) Expression of 1000 genes chosen at random. The genes are listed along the rows, while the samples are listed down the columns. The samples are laid here according to the chronological order of their development. The samples labeled T6 (256 cells, left) and T7 (Dome, right) are found on both sides of the green line, whereas the samples labeled T16 (hatching, left) and T17 are found on both sides of the red line (3 days after fertilization, right). The samples that were obtained before zygotic genome activation are found to the left of the green line, and the samples that were collected between hatching and the end of the study are found to the right of the red line (including juvenile fish and 9 tissues). The order of the samples in (**B**), (**C**), (**H**), and (**I**) and (**J**) is the same as the order of the samples in (**A**); (**B**) Expression of all transposons; (**C**) Expression of transposons of rnd-3_family-293; (**D**) The diagram displays the structural pattern of rnd-3_family-293; (**E**) The motif 5; (**F**) The motif 1; (**G**) The structure pattern displays the motifs annotated by MEME for the transposons of rnd-3_family-293 (25 were shown); (**H**) The expression of genes containing both motif 1 and motif 5; (**I**) The expression of genes containing only motif 1; (**J**) The expression of genes containing only motif 5.

**Figure 6 ijms-23-11547-f006:**
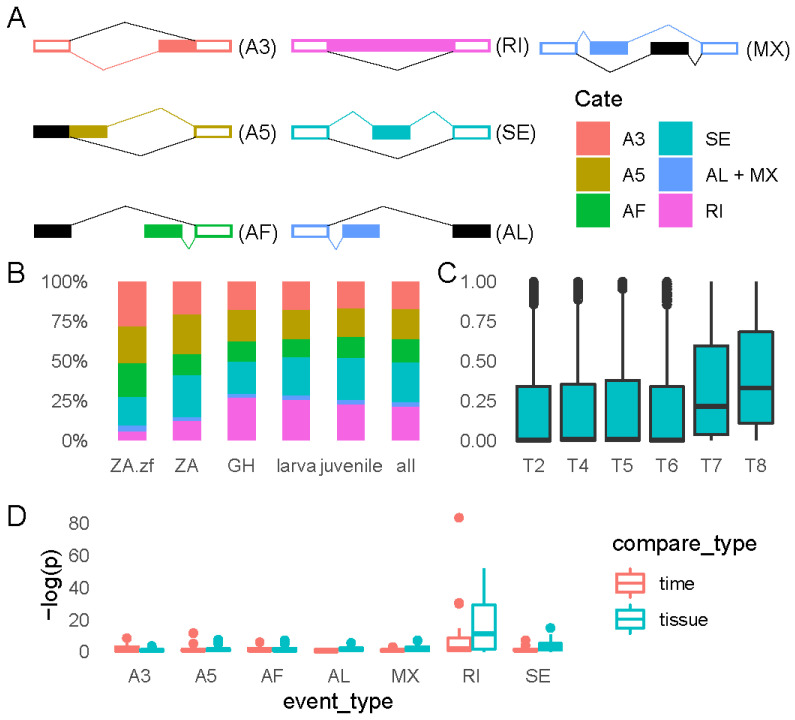
An examination of the various possibilities for splicing. (**A**) An example of a splicing pattern that has seven alternative splicing events that are fundamentally distinct from one another. In the diagram, the two distinct methods of splicing are indicated by the colors color and black, respectively. These colors represent the two possible outcomes of the process. The exons that are shared by both of the possible splicing types are denoted by the colored boxes that are otherwise empty. When the Ψ value is calculated, the colored splicing forms represent form 1; (**B**) The ratio of various alternative splicing events expressed at different stages. The ZA.zf indicates the ZGA stage of zebrafish [32]. Samples of grass carp were collected when the fish were in the ZGA stage (T2–T8), the GH stage (T9–T16), and the larva stage (T17–T20), respectively, as indicated by the letters ZA, GH, and larva, which are denoted by those letters. The statistical analysis was carried out on each and every sample of grass carp; (**C**) The variations in the values Ψ of alternative splicing events of the RI type that occurred during the ZA period; (**D**) A description of the distribution of the *p*-value. Classification of the comparison: tissue shows that the comparisons were carried out between 9 tissue samples pairwise and time indicates that the comparisons were carried out between adjacent time intervals in the time series samples from T2–T20.

**Figure 7 ijms-23-11547-f007:**
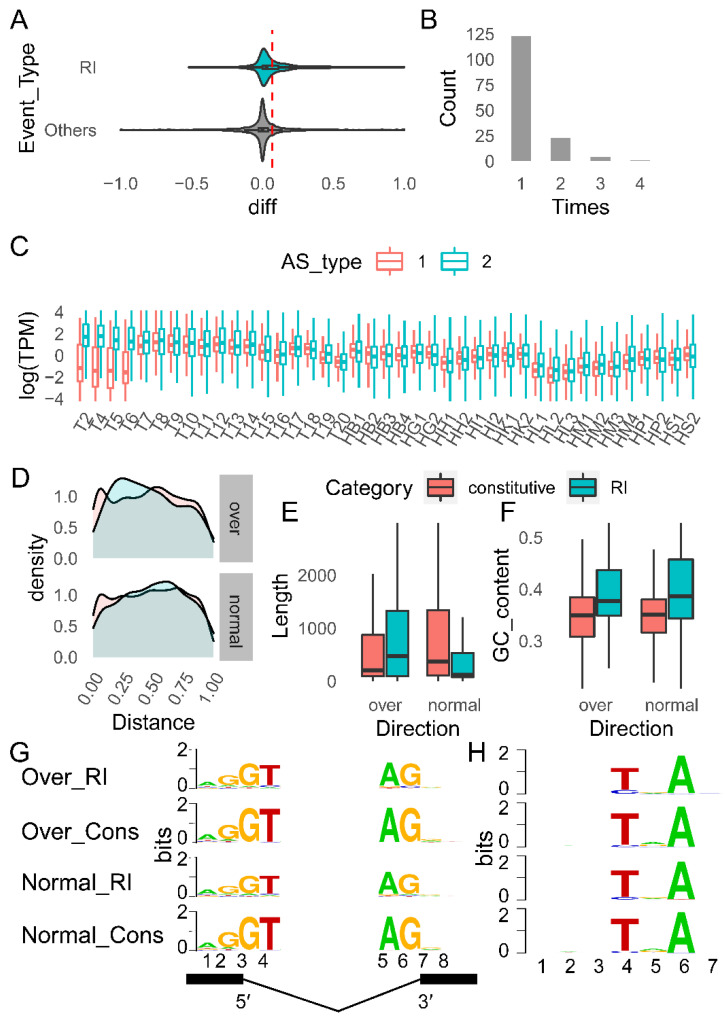
The distinguishing features of the transition from T6 (Dome) to T7 in the RI alternative splicing (256 cells). (**A**) The difference between the Ψ values of RI events (Ψ(T7)−Ψ(T6)) exhibited an obvious right-skewed distribution. The right side of the red dotted line (0.07), which was classified as an “over” event (which will be referred to as “over” later), and the left side of the line, which was defined as a “normal” event (which will be referred to as “normal” hereafter); (**B**) The total number of RI events that have taken place in a gene; (**C**) The expression ratio of the two different splicing forms that have taken place in RI events. The transcripts with the intron kept are denoted by the form “1,” while the transcripts with the intron spliced out are denoted by the form “2.”; (**D**) The position distribution of alternative introns (red) and constitutive introns (blue) on the gene (relative to the transcriptional start site); (**E**) The differences in GC content between alternative introns and constitutive introns; (**F**) The differences in length between alternative introns and constitutive introns; (**G**) The motif of the splice site. (**H**) Motifs of branch-point sequences are also referred to (BPS). Alternative introns of “over” transcripts are referred to as “Over_RI,” while constitutive introns of “over” transcripts are referred to as “Over_Cons.” Similarly, alternative introns of “normal” transcripts are referred to as “Normal_RI,” while constitutive introns of “normal” transcripts are referred to as “Normal_Cons”.

**Table 1 ijms-23-11547-t001:** Sample information.

Sample ID	Time after Fertilization	Tissue	Characteristics
T2	0 hpf	ovum	ovum
T4	1.6 hpf	embryo	1, 2, 4 cells
T5	2.8 hpf	embryo	64 cells
T6	3.7 hpf	embryo	256 cells
T7	5.2 hpf	embryo	dome
T8	6.3 hpf	embryo	50% epiboly
T9	7.9 hpf	embryo	75% epiboly
T10	9.1 hpf	embryo	90% epiboly ~ bud
T11	11.1 hpf	embryo	4~6 somite
T12	12.9 hpf	embryo	15 somite
T13	14.8 hpf	embryo	18~19 somite
T14	18.0 hpf	embryo	25~26 somite, starting to shrink
T15	21.8 hpf	embryo	rotation
T16	31.8 hpf	full fish	hatching
T17	3 dpf	full fish	yolk exhausted, appearance of eyes, start to swim
T18	6 dpf	full fish	first feeding
T19	13 dpf	full fish	NA
T20	20 dpf	full fish	NA
B1, M1	39 dpf	B, M	NA
B2, M2, L1	69 dpf	B, M, L	NA
B3, M3, L2, G1, Sk1, Sp1, K1, H1, I1	134 dpf	B, M, L, G, Sk, Sp, K, H, I	NA
B4, M4, L3, G2, Sk2, Sp2, K2, H2, I2	197 dpf	B, M, L, G, Sk, Sp, K, H, I	NA

Note: hpf, hours after fertilization; dpf, days after fertilization. Abbreviations for tissues are B—brain, M—muscle, L—liver, G—gill, Sk—skin, Sp—spleen, K—kidney, H—heart, and I—intestine.

**Table 2 ijms-23-11547-t002:** Statistics of expressed transposable elements.

Class	Count	Ratio in Expressed TE	Count in Genome	Fraction in Total Transposon Elements
unclassified LTR	2	3.13%	14,757	0.88%
DNA/TcMar	8	12.50%	106,014	6.29%
LINE/L2	13	20.31%	44,915	2.66%
unclassified SINE	41	64.06%	10,123	0.60%

Note: In the repeat element annotation of grass carp genome, “unclassified LTR” and “unclassified SINE” belong to long terminal repeats (LTRs) and short interspersed nuclear element (SINE), respectively. They were not assigned with a clear sub-class [54].

**Table 3 ijms-23-11547-t003:** Characteristics of alternative introns in RI (compared with constitutive introns).

Category	Over Events	Normal Events	Classic Features [80]
Relative position	5′ (not significant)	3′ (not significant)	3′
Length	Long	Short	Short
GC content	High	High	High
Strength of splice site	Weak	Weak	Weak
Strength of BPS	Weak	Weak	Weak

## Data Availability

The sample information and sequencing data have been submitted to the National Genomics Data Center (NGDC, https://ngdc.cncb.ac.cn/, accessed on 10 June 2022) with the accession number PRJCA010000.

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
