# Peer review of "Dynamic Transcriptional Landscape of Grass Carp (Ctenopharyngodon idella) Reveals Key Transcriptional Features Involved in Fish Development"

_ijms, 2022, doi:10.3390/ijms231911547_

Round 1

Reviewer 1 Report

This paper shows the transcriptome analysis throughout development and tissues of grass carp. This is a very comprehensive analysis and such data are very valuable to fish community as well as evolutionary and comparative studies. Their analysis reveals that time specific expression of transposons and time- and tisssue-specific expression of alternative splicing differences. These findings suggest important roles for these genes and these splice variants. Especially, this wide and deep transcriptome analysis reveals that expression of specific splice variant correlates with MBT or full activation of zygotic genome. These findings are very interesting and is expected to contribute greatly to the future development of this field. I recommend to accept this paper with minor revisions.

1. I think RI in abstract should be spelled out.

2. The figure legends in supplementary figures should be more detailed. Especially Supplementary Figure S10.

Author Response

1. I think RI in abstract should be spelled out.
R: We have supplemented the full spelling of RI in the abstract as you suggested.

2. The figure legends in supplementary figures should be more detailed. Especially Supplementary Figure S10.
R: We have added more details in the legend of Supplementary Figure S10, as well as other supplementary figures. 

Reviewer 2 Report

In this manuscript [title ‘ Dynamic Transcriptional Landscape of Grass Carp (Ctenopha-ryngodon idella) Reveals Key Transcriptional Features Involved in Fish Development' ] Xia and co-workers provide a high-quality reference transcriptome and dynamic transcriptional map of grass carp. In this study the authors analyzed the transcriptional landscape during grass carp development, deeply investigating the changes of lncRNAs expression and alternative splicing events, in this phenomenon. Furthermore, they identified a novel family of short interspersed nuclear element (SINE), rnd-3 family-293, associated with zygotic genome activation.   

Overall, the manuscript is of general interest and adds new knowledge in the field.  

However, the authors should address comment before publication. 

-       In the Table1, column “Sample ID” the authors should specify what the acronyms “H(B/M)1; H(B/M)2…..” means.

-       Authors should correct the figure and table numbers in order in which they are stated along the text. In line 123, the authors mention Supplementary Table S3, while Supplementary Table S1 and S2 are mentioned in Material and Method section at the end of the manuscript. This happens also for some figure panels. Please correct this to make the read of the manuscript easier. 

-       In line 157, only Figure 2A is mentioned, but It is not clear why Figure 2B is not mentioned; it is present as a separate figure panel. The authors should also make one figure 2 (without panels A and B) and refer to Figure 2A and Figure 2B as: Figure 2A, top and bottom panels.

-       In Line 192-196 Figure 4B is not mentioned. To make the read easier, I suggest to uniform figure with the text of the manuscript.   

-       Figure 5E should be integrated in the figure 5A-B-C-I-J-K, since it is an element that the reads use to understand them.

-        

Author Response

-       In the Table1, column “Sample ID” the authors should specify what the acronyms “H(B/M)1; H(B/M)2…..” means.
R: To make the column "Sample ID" clearly, we have listed the IDs for all tissue samples now.

-       Authors should correct the figure and table numbers in order in which they are stated along the text. In line 123, the authors mention Supplementary Table S3, while Supplementary Table S1 and S2 are mentioned in Material and Method section at the end of the manuscript. This happens also for some figure panels. Please correct this to make the read of the manuscript easier. 
R: We have renamed the (supplementary) figures and tables according to the order they were cited.

-       In line 157, only Figure 2A is mentioned, but It is not clear why Figure 2B is not mentioned; it is present as a separate figure panel. The authors should also make one figure 2 (without panels A and B) and refer to Figure 2A and Figure 2B as: Figure 2A, top and bottom panels.
R: We have modified Figure 2 as your suggested.

-       In Line 192-196 Figure 4B is not mentioned. To make the read easier, I suggest to uniform figure with the text of the manuscript. 
R: Yes, Figure 4B was first mentioned on line 207, and it is the last panel of Figure 4 referred in main text. Therefore, we have swap the names of Figure 4B and Figure 4D, and uniformed this figure with the manuscript text as you suggested.

-       Figure 5E should be integrated in the figure 5A-B-C-I-J-K, since it is an element that the reads use to understand them.
R: We have integrated Figure 5E in the Figure 5A-B-C-H-I-J, and rearranged the panels of Figure 5.